# Impact of the Gut Microbiota Balance on the Health–Disease Relationship: The Importance of Consuming Probiotics and Prebiotics

**DOI:** 10.3390/foods10061261

**Published:** 2021-06-02

**Authors:** Laura-Berenice Olvera-Rosales, Alma-Elizabeth Cruz-Guerrero, Esther Ramírez-Moreno, Aurora Quintero-Lira, Elizabeth Contreras-López, Judith Jaimez-Ordaz, Araceli Castañeda-Ovando, Javier Añorve-Morga, Zuli-Guadalupe Calderón-Ramos, José Arias-Rico, Luis-Guillermo González-Olivares

**Affiliations:** 1Área Académica de Química, Instituto de Ciencias Básicas e Ingeniería, Universidad Autónoma del Estado de Hidalgo, Mineral de la Reforma 42184, Hidalgo, Mexico; ol232998@uaeh.edu.mx (L.-B.O.-R.); elizac@uaeh.edu.mx (E.C.-L.); jjaimez@uaeh.edu.mx (J.J.-O.); ovandoa@uaeh.edu.mx (A.C.-O.); anorvej@uaeh.edu.mx (J.A.-M.); 2Departamento de Biotecnología, División de Ciencias Biológicas y de la Salud, Universidad Autónoma Metropolitana, Unidad Iztapalapa, Av. San Rafael Atlixco 186, Ciudad de Mexico 09340, Mexico; 3Área Académica de Nutrición, Instituto de Ciencias de la Salud, Universidad Autónoma del Estado de Hidalgo, Circuito Ex Hacienda, La Concepción S/N, Carretera Pachuca Actopan, San Agustín Tlaxiaca 42060, Hidalgo, Mexico; esther_ramirez@uaeh.edu.mx (E.R.-M.); zramos@uaeh.edu.mx (Z.-G.C.-R.); 4Área Académica de Ingeniería Agroindustrial e Ingeniería en alimentos, Instituto de Ciencias Agropecuarias, Universidad Autónoma del Estado de Hidalgo, Av. Universidad km. 1, Ex-Hacienda de Aquetzalpa, Tulancingo 43600, Hidalgo, Mexico; aurora_quintero1489@uaeh.edu.mx; 5Área Académica de Enfermería, Instituto de Ciencias de la Salud, Universidad Autónoma del Estado de Hidalgo, Circuito Ex Hacienda, La Concepción S/N, Carretera Pachuca Actopan, San Agustín Tlaxiaca 42060, Hidalgo, Mexico; josearias.rico@hotmail.com

**Keywords:** microbiota, microbiota–intestine–brain axis, probiotic, prebiotic

## Abstract

Gut microbiota is a group of microorganisms that are deposited throughout the entire gastrointestinal tract. Currently, thanks to genomic tools, studies of gut microbiota have pointed towards the understanding of the metabolism of important bacteria that are not cultivable and their relationship with human homeostasis. Alterations in the composition of gut microbiota could explain, at least in part, some epidemics, such as diabetes and obesity. Likewise, dysbiosis has been associated with gastrointestinal disorders, neurodegenerative diseases, and even cancer. That is why several studies have recently been focused on the direct relationship that these types of conditions have with the specific composition of gut microbiota, as in the case of the microbiota–intestine–brain axis. In the same way, the control of microbiota is related to the diet. Therefore, this review highlights the importance of gut microbiota, from its composition to its relationship with the human health–disease condition, as well as emphasizes the effect of probiotic and prebiotic consumption on the balance of its composition.

## 1. Introduction

The gut microbiota is a group of microorganisms that colonize the gastrointestinal tract, and is found in a higher proportion than the cells of the human body [1]. About half of the fecal mass constitutes the microorganisms that are essentially grouped into five phyla: Firmicutes, Bacteroidota, Verrucomicrobia, Actinobacteria, and Proteobacteria, with a 90% predominance of the first two [2]. Due to the diversity of microorganisms that makes them the most important environmental agent in the human body, various studies indicate that gut microbiota is directly associated with both the health of the host and some diseases [3]. Despite the evidence on the relationship of gut microbiota with the parameters of various diseases, more research is needed to evaluate other factors, such as the interaction between host genetics, diet, and metabolism regulation [4]. 

Once the microbiota is established in an individual, it usually changes in a short time, and the changes are even generated in stages during the life of the human. Notwithstanding its high variability, the alteration of gut microbiota composition has huge implications in the pathogenesis of a wide range of diseases, from chronic gastrointestinal diseases to neurological disorders. The influence of gut microbiota in the development of diseases is so great that some investigations have shown alterations in the serotonergic neurotransmission of the central nervous system (CNS), which were secondary to gut microbiota imbalance (dysbiosis) [5]. For this reason, enormous efforts have been directed to reverse the effect of intestinal dysbiosis in neurodegenerative diseases. 

One of the main factors affecting the concentration changes over specific microorganisms of the gut microbiota is the combination of diet with genetic factors [6]. It explains why, in addition to gastrointestinal diseases, the microbiota has a direct effect on the development of diseases such as type 2 diabetes mellitus (DM2) and obesity, which are part of the so-called metabolic syndrome [7]. Although it has been suggested that an adequate balance between Firmicutes and Bacteroidota is necessary to avoid the appearance of this type of disease, the studies are not conclusive in this regard [8]. However, gut dysbiosis is a preponderant factor in its development and control. Additionally, dysbiosis has been presented as a risk factor in the development of various types of cancer, and, in addition to the genetic factor, abnormal microbial translocation and molecular mimicry are directly related to this condition [9]. Even though the microorganisms of gut microbiota do not directly induce tumorigenesis, they could interact with the immune system to indirectly promote the proliferation of cancer cells [10].

That is why, knowing that diet has a significant impact on the establishment and composition of gut microbiota throughout life [11], this review aims to provide information about the relationship between the consumption of probiotics and prebiotics and the establishment of balanced gut microbiota. Moreover, the generalities of the microbiota and its importance in the establishment and development of chronic and degenerative diseases are disclosed. The relationship between the gut microbiota–brain axis, its importance in the development of psychiatric and neurodegenerative diseases, as well as its association with the development of brain cancer, have also been addressed. Finally, the use of fermentable oligosaccharides, disaccharides, monosaccharides, and polyols (FODMAPs) and their impact on the generation of beneficial populations in the digestive system, has also been mentioned in this review. 

## 2. Human Gut Microbiota: Importance and Composition

The gut microbiota is a group of organisms that provides various benefits and imparts resistance to the colonization of new species, maintaining a symbiotic relationship with the host. However, an imbalance in this complex community could lead to recolonization by pathogenic microorganisms, causing inflammatory processes and the evolution of various diseases [12]. This suggests that gut microbiota maintains the homeostasis of the human intestine [13]. It provides various effects such as protection against pathogens, carbohydrate digestion, regulation of fat storage, production of essential vitamins, and modulation of the immune response, representing an environmental factor of great importance in human homeostasis [14].

The gut microbiota has prominent participation in various metabolic functions through the fermentation of indigestible carbohydrates and the production of short-chain fatty acids (SCFAs), with beneficial systemic effects [15]. Among the SCFAs, acetate, propionate, and butyrate are the most frequently produced [16]. These compounds appear to regulate mechanisms such as lipogenesis and cholesterol biosynthesis in the liver. Acetate is the SCFA achieving the highest concentration in plasma, which has been related to low plasma insulin levels, propionate contributes to gluconeogenesis in the liver, and butyrate is used mainly as an energy source by the gut mucosa [17]. Additionally, gut microbiota allows the production of ketone bodies and carbon dioxide, as well as the regulation of energy homeostasis by stimulating intestinal enteroendocrine cells [17,18,19]. 

In addition to metabolic functions, gut microbiota contributes to the prevention of colonization of pathogenic microorganisms. This activity involves the production of bacteriocins or antimicrobial peptides [20]. Consequently, there is a competition for nutrients, which stimulates innate immunity through the secretion of IgA [21,22] and the activation of Toll-like cell receptors (TLCR). These compounds are capable of identifying molecular patterns associated with microorganisms that structurally include lipopolysaccharide of bacterial origin (LPS), lipoprotein, flagellin, and DNA of pathogens [15]. That is why the immunomodulatory activity exerted by the intestinal microbiota is involved in the interaction with cells of the immune system, participating in both the stimulation of innate immunity and the maturation and subsequent development of adaptive immunity [23,24].

Finally, the activity performed by the intestinal microbiota at the neurological level is crucial. This is carried out through bidirectional communication between the intestine and the brain through the enteric nervous system (ENS) [25]. Gut microbiota controls the ENS through the production, expression, and turnover of neurotransmitters and neurotrophic factors, the maintenance of the sensory barrier, the modulation of enteric sensory input, the production of bacterial metabolites, and the immune regulation of the mucosa [15,26]. Additionally, evidence indicates an association between mood disorders and dysbiosis [27]. This is due to the impact of neurotransmitters, such as serotonin and dopamine, originated by native gut microbiota, on brain alertness, mood control, memory, and the learning process of an individual [26,28]. 

Certainly, the success of the activities carried out by the intestinal microbiota depends to a great extent on its composition, which changes over time according to the different stages of an individual’s life. This microbiota begins to appear from gestation and continues to develop in parallel with the host, fulfilling the necessary functions for the maintenance of homeostasis.

### Gut Microbiota and Aging

Approximately 10^6^–10^14^ microorganisms belonging to the domains eukaryotic, archaea, and bacteria colonize the human gastrointestinal tract. Gut microbiota is composed of approximately 1000 species, and the main phyla are Firmicutes and Bacteroidota, and, to a lesser extent, Fusobacteria, Cyanobacteria, Proteobacteria, Verrumicrobia, and Actinobacteria can also be found [29]. However, this composition undergoes changes over time (Figure 1) and the dominant bacterial phyla are different at each stage of human life [2]. 

Intestinal colonization begins in the gestational stage. Some studies have been able to determine the presence of various microorganisms in the placenta, umbilical cord, and amniotic fluid, Proteobacteria and Actinobacteria being the most predominant at this stage [30,31,32]. Later, during natural childbirth, newborns are colonized by taxa that originate in the mother’s vagina, while newborns delivered by cesarean section will be colonized by microorganisms present on the skin [33]. After birth, the intestine is progressively colonized by various microbial strains. The first colonizing microorganisms generally belong to the enterococci and enterobacteria, followed by members of strict anaerobes genera, such as *Bifidobacterium*, *Clostridium*, and *Bacteroides* [34].

By the age of three, the microbiota has stabilized. However, bacterial strains will undergo significant fluctuations and changes over time, modifying their composition and gene expression. These modifications are due to anatomical, dietary, nutritional, and environmental alterations [6]. Additionally, the composition will also be determined by pathological disorders, such as gastrointestinal and systemic infections, as well as by the use of pharmacological agents, such as antibiotics, laxatives, prokinetics, and probiotics [35,36].

During older age, a restructuring in the intestinal microbiota has been observed. Claesson et al. [37] reported significant changes in subjects older than 65 years, specifically an increase in the abundance of Bacteroidota and Proteobacteria. It has been hypothesized that alterations in the microbiota upon reaching an older age are mainly due to physiological changes in the gastrointestinal tract. Among these changes are the decrease in esophageal contractions and peristaltic movements, alteration in the gastric lining and fibrosis, the presence of low-grade chronic inflammation, and eating habits [38].

## 3. Role of Gut Microbiota in Human Health

### 3.1. Diabetes and Obesity (Metabolic Syndrome)

Diseases such as obesity and DM2, and, in general, metabolic syndrome are related to dysbiosis [7,35,39]. There are three mechanisms related to the microbiota and the development of these diseases. One of them is the type of carbon source for obtaining energy. The second is related to the modulation of some human genes and proteins, which are involved in the regulation of energy expenditure. Finally, the third mechanism is associated with the regulation of the levels of LPS of bacterial origin into the plasma, which can induce chronic subclinical inflammation. The last one leads to the development of insulin resistance through the activation of TLCR [40,41]. 

Most human studies indicate that an increase in the Firmicutes/Bacteroidota ratio is related to an increase in a low-grade inflammation state. Additionally, the little diversification of the intestinal microbiota is also associated with greater insulin resistance, inflammation, and adiposity [42,43]. This information is reinforced with metagenomic studies of the human intestinal microbiota relating not only obesity and insulin resistance, but also the increase of various markers, including TNF-α interleukin 6, and other proinflammatory cytokines [44,45,46].

Thus, the role of the intestinal microbiota in obesity and diabetes has also been demonstrated during the transplantation of fecal microbiota from lean human donors to obese human recipients [47]. Microbiota transplantation in these conditions increases insulin sensitivity and, therefore, there is better control of glycemic levels in subjects with metabolic syndrome [48]. Furthermore, a significant reduction in body weight in overweight and obese subjects is achieved through treatment with probiotics that could colonize the gastrointestinal tract, as demonstrated by Kadooka et al. [49] by supplying *Lactobacillus gasseri* as a supplement.

In the specific case of diabetes, some studies have indicated changes in the composition or function of the intestinal microbiota in patients with this condition. It is known that dysbiosis in this pathology is related to a decreased population of butyrate-producing bacteria, such as *Faecalibacterium prausnitzii* and *Roseburia intestinalis* [50]. The intestinal production of SCFAs, such as butyrate, is related to the beneficial effect on peripheral tissues, such as the liver, and adipose and connective tissue, also improving insulin sensitivity [51]. In addition, there is an increase in the populations of opportunistic pathogens, such as *Escherichia coli*, *Bacteroides caccae*, *Erysipelatoclostridium ramosum*, *Clostridium symbiosum*, and *Clostridium hathewayi*, which have also been related to dysbiosis [52].

Metagenomic studies have revealed a direct relationship between dysbiosis and diseases such as DM2 and obesity, since a decrease in the population of butyrate-producing species of the genera *Clostridium*, *Fecalibacteria*, and *Roseburia*, belonging to the phylum Firmicutes, has been demonstrated [35,44,53]. Furthermore, the Firmicutes/Bacterioidetes relationship is altered by the increase not only of *Escherichia coli*, but also of the phylum in general [35,54,55]. These results demonstrate a link between microbiota and metabolic diseases such as obesity and diabetes, and pose an area of opportunity for the development of new therapeutic strategies.

### 3.2. Gastrointestinal Diseases

Despite the mutualistic relationship that exists between the microorganisms of the gut microbiota and the human, some bacteria can acquire virulence and change their symbiotic properties due to genetic, environmental, and dietary factors [56,57]. Various studies suggest that the alteration of gut microbiota, as well as its metabolic functions, are correlated with the appearance and progression of gastrointestinal diseases, such as severe diarrhea, celiac disease, and irritable bowel syndrome, among others [58,59,60].

Thus, the evolution of diseases such as intestinal inflammation is governed by complex interactions between several factors: environmental risks, host genetics, and the state of gut microbiota [61,62]. However, it is gut microbiota that has a direct effect when there is a reduction in the population of Firmicutes [56]. This fact is well documented through clinical studies that show that the diversity and richness of the microbiota are significantly reduced in patients with intestinal inflammation [56,58,63]. Furthermore, the pathogenesis of this disease is characterized by the accumulation of certain pathobionts, such as *Escherichia coli* and *Ruminococcus gnavus* [64,65]. 

On the other hand, in ulcerative colitis, which is characterized by inflammation and ulceration of the lining of the colon, an adaptation of microbial species such as *Bacteroides fragilis* and *Escherichia coli* has been observed. These microorganisms have evolved to acquire adherence to the mucosa of the ileum and degrade its walls [66,67,68]. Similarly, the presence of pathogenic microorganisms adhered to the intestinal wall of patients with irritable bowel syndrome has been manifested [67,69,70]. Furthermore, there is an increase in the Firmicutes/Bacteroidota ratio compared to that of healthy patients [70,71]. Specifically, there is a greater number of species of the family Ruminococcaceae and *Clostridium cluster* XIVa and a smaller number of *Bacteroides* [72].

In certain gastrointestinal diseases, dysbiosis has been related to genetic mutations [73]. For example, in the case of celiac disease, it has been reported that mutations present in genes involved in the secretion of intestinal mucus, the structure of associated bacteria, and a reduction in bacterial diversity and richness, influence the pathogenesis of the disease [74]. This disease presents a series of symptoms of chronic immune-mediated inflammation in the small intestine due to a lower abundance of *Bifidobacterium* spp. in the human intestine [75,76]. This marks an association with an increased risk for the development of autoimmune diseases [57].

More in-depth studies are required to explore the therapeutic potential of the modulation of gut microbiota in the treatment of gastrointestinal diseases since, due to scientific evidence, the phenotype of commensal bacteria can go from symbiotic to pathogenic in response to various risks factors [56]. These phenotypic alterations modify not only the host’s immune system, but also impact the structure and diversity of gut microbiota, which leads to the development and/or progression of a greater diversity of gastrointestinal diseases [77,78,79].

### 3.3. Psychiatric and Neurodegenerative Diseases

The microbiota–intestine–brain axis (Figure 2) is a bidirectional communication network that includes the central nervous (CNS), autonomic nervous (ANS), and enteric nervous (ENS) systems. The immune system, the endocrine system, and the intestinal microbiota also belong to this axis [80]. Given the complexity of this network, possible intervention strategies have been explored, aimed at the dysbiosis of gut microbiota present in various neurological and psychiatric disorders, including the use of probiotic, prebiotic, and symbiotic foods [2].

Some studies have pointed out that gut microbiota dysbiosis also plays an important role in psychiatric and neurological diseases, such as Alzheimer’s, Parkinson’s, autism, neurodegeneration, multiple sclerosis, anxiety, and depression [81,82]. Furthermore, various studies indicate that the intestinal microbiota influences the gut–brain system, triggering the symptoms that occur during a state of anxiety and stress [83]. Likewise, the intestinal microbiota seems to have an impact on pain tolerance mechanisms [84,85]. Similarly, it has been noted that gut microbiota is closely related to neurological functions, mood, and host behavior, as well as the circadian cycle [81]. However, the mechanisms by which this relationship is generated are still not entirely clear, but the concept of the “microbiota–gut–brain axis” has been intended to explain them [80]. Some of these mechanisms are as follows:(1)Involvement of the vagus nerve. There is a connection between the ENS and the CNS that provides a direct communication pathway between gut microbiota and the CNS [86,87];(2)Participation of the circulatory system. This system regulates the effects of various metabolites, such as neurotransmitters, hormones, and SCFA, that are produced by gut microbiota and impact on CNS functions [81];(3)Regulation of signals and the synthesis of neurotransmitters. Gut microbiota apparently modulates the expression of central neurotransmitters and related receptors, and some species produce neurotransmitters, such as acetylcholine, dopamine, and adrenaline, or induce their synthesis [88,89,90];(4)Production of SCFA. Gut microbiota is capable of modulating the maturation of the microglia and the permeability of the blood–brain barrier through the synthesis of SCFA [89,91,92];(5)Immunomodulation. Gut microbiota influences the activation of peripheral immune cells that regulate CNS immune reactions [93,94].

Studies have shown differences between patients with neurological disorders and healthy controls [95]. In healthy individuals, an analysis of 16s rRNA from fecal microbiota showed that bacteria corresponding to the phyla Firmicutes and Bacteroidota have a higher proportion than that of the phyla Proteobacteria, Actinobacteria, Fusobacteria, and Verrucomicrobia [96]. Additionally, microbial abundance and diversity are significantly reduced in patients with depression and anxiety disorders [95]. Thus, the concentration of the Lachnospiraceae and Ruminococcaceae families and the *Ruminococcus* and *Lactobacillus* genera is decreased [97]. This dysbiosis causes both microorganisms and the products of their metabolism to induce inflammation at the brain level through blood circulation and induce the production of various cytokines, such as IL-6, IL-1β, and TNF-α [95], which, in turn, modulate various brain processes that affect mood and behavior [98].

In the case of Parkinson’s patients, gut microbiota alterations coincide with an aggravation of the condition [99,100,101,102], which is related to a lower concentration of *Prevotellaceace* species compared to the relative abundance of *Enterobacteriaceae* in the feces of these patients [103]. Likewise, a decrease in the *Prevotellaceae* population generates an increase in intestinal permeability and systemic exposure to LPS [87,104]. This endotoxin induces systemic inflammation by the production of proinflammatory cytokines that interact with TLCR and nuclear factor-kappa B (NF-κB). This mechanism is related to the progression of Parkinson’s once LPS breaks the blood–brain barrier [105,106].

Another condition related to dysbiosis is Alzheimer’s, a neurodegenerative disorder that leads to cognitive dysfunction [107]. An increase in intestinal permeability, as a consequence of dysbiosis, has, in turn, been associated with an increase in the concentration of various microorganisms, as well as products derived from their metabolism [108,109,110]. The released endotoxins are involved in neuroinflammation in patients with Alzheimer’s by acting on the innate immune system [111,112]. In general, it has been observed that gut microbiota in people with this condition is represented by an increase in Proteobacteria and a decrease in SCFA-producing bacteria [113]. This imbalance leads to an increase in proinflammatory cytokines, such as TNF-α, IL-5, IL-6, IL-1β, and IL-8 [114]. Thus, the use of probiotics has been explored as an alternative to improve the cognitive functions of patients and the decrease in cytokines that lead to neuroinflammation [115,116,117].

### 3.4. Cancer

The role of the gut microbiota in the development of several types of cancer has been reported in recent years [118,119,120]. It has been highlighted that pathogenesis is not only attributed to genetic susceptibilities, but also to mechanisms that include abnormal microbial translocation, molecular mimicry, and dysregulation of local and systemic immunity [9]. In this context, it has been reported that some microorganisms belonging to gut microbiota have oncogenic effects or oncolytic activity in tumor cells. Approximately 20% of cancers are attributable to infectious agents, including bacteria [121]. In addition to this, it has been observed that there is a difference between healthy individuals and cancer patients in terms of population and microbial diversity present at the intestinal level [122]. 

The link between gut microbiota and carcinogenesis has been described, with an emphasis on bacterial metabolites. Thus, the main mechanisms of bacterial-mediated carcinogenesis are mainly based on the effects of the specific toxins or virulence factors produced [123,124]. Furthermore, microbial metabolites, such as polyamines and secondary bile acids, are also involved in cancer cell proliferation and tumor induction through the β-catenin signaling pathway [125], epidermal growth factor receptor (EGFR) transactivation [126], and increased COX-2 activity [127].

That is why the immune system of the intestinal mucosa is related to cancer, in which its interaction with gut microbiota is considered a key factor in the maintenance of homeostasis [68,128]. Due to this interaction, there is an effect on the inhibition of bacterial adhesion and colonization, as well as the induction of cell differentiation [129]. In this sense, microbial species such as *Bacteroides fragilis* induce the differentiation of T cells CD4+ to regulatory T cells (Treg cells) [130], which are capable of secreting large amounts of anti-inflammatory cytokines, such as IL-10, and recognizing antigenic substances associated with the bacterial genera *Clostridium* and *Bacteroides* [131,132]. However, the intestinal microbiota not only has an immunomodulatory effect at the local level, but also the systemic level [128]. Metabolites produced by microorganisms enter the bloodstream and, therefore, affect the immune response in distant organs through interaction with TLCR [129]. 

Although the microorganisms of gut microbiota do not directly induce tumorigenesis, they could interact with the immune system to indirectly promote the proliferation of cancer cells [10]. Thus, a defective immune response increases the abundance of certain bacterial genera and triggers signaling pathways that lead to the transcription of oncogenes [10,128]. In addition to this, gut microbiota can indirectly promote cancer by inducing inflammation or immunosuppression through the production of cytokines [133,134]. Finally, the development of various malignant neoplasms, including some types of cancer (gastric, colorectal, pancreatic, breast, and brain cancer), is currently associated with variations in gut microbiota composition [135].

#### 3.4.1. Colorectal Cancer

Some bacterial species related to the development of colorectal cancer, such as *Fusobacterium nucleatum*, *Peptostreptococcus anaerobius*, and *Bacteroides fragilis* enterotoxigenic, have been identified [136]. Moreover, a lower probability of survival in patients with colon cancer has been associated with *F. nucleatum* abundance [137] due to the induction of chemoresistance, which activates autophagy [134], leading to treatment failure or disease recurrence [128]. The species mentioned above induce tumor proliferation [138], promote inflammation [139], protect the tumor from the mechanisms exerted by the immune system [140], and cause damage to host cell DNA [141]. All these factors contribute to carcinogenesis. As in other types of cancer, in colorectal cancer, special emphasis has been placed on protein toxins produced by the intestinal microbiota [142]. The procarcinogenic effect of these toxins could be due to the direct attack on DNA, which leads to genomic instability or proliferation and induction of resistance to apoptosis in cancer derived from cellular signaling alterations [143].

#### 3.4.2. Pancreatic Ductal Adenocarcinoma

This condition represents one of the most serious malignant neoplasms, with overall survival lower than 5 years [9]. Because surgical resection is often not possible, treatment is focused on chemotherapy. However, some patients could develop chemoresistance [144] associated with gut microbiota, which has a major impact on pancreatitis and pancreatic ductal adenocarcinoma [144,145].

In the same way that *Fusobacterium nucleatum* has been associated with colorectal cancer, it is currently known that this microorganism induces chemoresistance, autophagy, and inflammation in pancreatic carcinogenesis processes [134,146]. Additionally, in patients with this condition, an increase in Proteobacteria and Verrucomicrobia, and a decrease in Firmicutes and Bacteroidota have been observed, which are accompanied by the activation of inflammatory pathways in tumor tissues [147].

Furthermore, it has been observed that the presence of intratumoral pathogens and bacteria, such as *Acinetobacter*, *Aquabacterium*, *Oceanobacillus*, and *Rahnella*, are associated with a higher risk of presenting pancreatic ductal adenocarcinoma [148,149]. The development of this disease involves the intestinal mucosa, epithelial and dendritic cells (DC), and different cells from the immune system [150]. The above-mentioned microorganisms are part of the gut microbiota and promote the development of adenocarcinoma through the release of a large number of metabolites [133] which interact with TLCR and also induce systemic inflammation and immune responses associated with pancreatic carcinogenesis and therapeutic resistance [151].

#### 3.4.3. Breast Cancer

The most common type of cancer affecting women worldwide is breast cancer. More than 40,000 deaths per year occur, even though there has been significant progress in its diagnosis and treatment [152]. A strong link between dysbiosis and the appearance of neoplasms, including those of breast cancer, has been shown [124]. Thus, recent research has focused on the influence of gut microbiota on the development of breast cancer, beyond genetic, environmental, and lifestyle factors [153]. 

The differences in the concentration of *Bifidobacterium*, *Faecalibacterium prausnitzii*, and *Blautia* have been used as biomarkers associated with the clinical stage of breast tumors [154]. In addition, these differences are also associated with the body mass index of the patients. Indeed, it has been observed that overweight and obese women with breast tumors present lower concentrations of Firmicutes, *Faecalibacterium prausnitzii*, and *Blautia* spp., as well as *Akkermansia muciniphila* prevalence, compared to patients with normal weight [154,155].

Likewise, in postmenopausal women with breast cancer, there is an alteration in the composition of the fecal microbiota, as well as a lower microbial diversity [156]. Specifically, elevated levels of Clostridiaceae, Faecalibacterium, and Ruminococcaceae, as well as a decrease in the proportion of Dorea and Lachnospiraceae, have been reported [157]. Similarly, an increase in the population of species such as *Escherichia coli*, *Citrobacter koseri*, *Acinetobacter radioresistens*, *Salmonella enterica*, and *Fusobacterium nucleatum*, among others, has been observed [158]. However, some factors must be taken into account when determining the characteristics of the gut microbiota of patients with breast cancer, such as age, ethnicity, and geographic location [156].

#### 3.4.4. Gastric Cancer

Gastric cancer is one of the most common neoplasms, which is characterized by acute and persistent inflammation [159]. In the same way, as in other types of cancer, the gut microbiota is related to the development of this disease and *Helicobacter pylori* is the main carcinogenic agent [160]. The mechanism of action of *H. pylori* to produce inflammation is associated with the degree of specific virulence of each strain [161]. The carcinogenesis process begins with genetic instability caused by the breaking of the host’s DNA chain [162]. Similarly, the TLCR and NOD-like receptors that recognize the presence of *H. pylori* are also associated with the chronic carcinogenesis process [162]. 

Current data indicate that *H. pylori* infection produces a genotoxic effect through two possible mechanisms (Figure 3) [163]. First, immune cell infiltration, including neutrophils and macrophages, increases, leading to the production of reactive oxygen and nitrogen species (RONS) [164]. RONS cause damage in the DNA leading to single-strand breaks and increased expression of oncogenes [165]. Alternatively, the transcription factor NF-κB is activated by RONS, inducing the expression of oncogenes and cell cycle regulators [166]. In addition, this factor translocates to the nucleus, forming an NER protein complex (XPG and XPF) which cleaves the promoter regions of genes, impacting gene expression because of double-strand breaks [167].

Although gastric acidity serves as an important barrier that limits the entry of microbes into the gastrointestinal tract [159], *H. pylori* can survive the conditions present in the stomach, but its ability to colonize gastric glands is restricted due to the large amount of acid produced in these cavities [168]. However, concomitant inflammation and the presence of *H. pylori* increases damage in various regions of the stomach. The result of this damage is atrophy associated with the abundance of this microorganism, which is greater in gastric cancer than in gastritis and intestinal metaplasia [169,170].

Additionally, in subjects with *H. pylori* infection and precancerous gastric lesions, variations in the relative abundance of the dominant phyla, such as Bacteroidota, Firmicutes, and Proteobacteria, have been observed in fecal microbiota [171]. Other specific bacteria related to gastric carcinogenesis, such as *Peptostreptococcus stomatis*, *Slackia exigua*, *Parvimonas micra*, *Streptococcus anginosus*, and *Dialister pneumosintes*, have been studied, but *H. pylori* is the microorganism commonly associated with gastric cancer [172].

#### 3.4.5. Brain Cancer

Gut microbiota and brain cancer association is a new topic that has gained interest in recent years [173]. This relationship can be explained by the mechanisms present in the microbiota–gut–brain axis, since it has been reported that they could influence the development or suppression of brain tumors [94].

Tryptophan is a substrate used by intestinal microorganisms to produce indoles. These molecules are involved in the signaling pathways between the gastrointestinal tract and the immune system [174]. This amino acid is metabolized in the kynurenine pathway, resulting in the biosynthesis of nicotinamide adenine dinucleotide and various neuroactive intermediates [173]. In this context, it has been reported that a dysregulation of the kynurenine pathway could contribute to the development of brain cancer by interrupting the antitumor immune response [175,176]. Likewise, gut microbiota could influence the brain tumor microenvironment through different mechanisms: (1) control of the expansion and activation of T cells [177]; (2) microglia [178,179,180]; (3) cytokine and arginine production, and tryptophan availability via kynurenine [179,181]; and (4) production of reactive oxygen species (ROS) and generation of antioxidants [119,180].

## 4. Modulation of Gut Microbiota through Diet

Diet is a preponderant factor that affects the establishment and composition of the gut microbiota throughout life [11], and changes during adulthood could affect intestinal homeostasis [182]. When there is a reduced dietary diversity and a lack of essential nutrients, a dysbiosis of gut microbiota occurs, leading to the appearance of various disorders [183]. The species that comprise the human gut microbiota require a wide range of nutrients and energy sources to promote growth, and they have a direct relationship with the effects associated with human health [182].

It has been highlighted that the intake of specific nutritional elements (carbon sources, nitrogen sources, growth factors, etc.) contributes to the diversification in the composition of the intestinal microbiota [184]. In this way, auxotrophies of some microorganisms that maintain the balance of the intestinal microbiota have been determined [185,186]. Marcobal et al. [187] point out that *Bifidobacterium infantis* and *Bacteroides thetaiotaomicron*, which are present in the gut microbiota of infants and adults, require oligosaccharides present in milk and the mucosa of the large intestine. This suggests that the particular carbon metabolism of each of the phyla plays a preponderant role in both the survival and long-term stability of gut microbiota.

The amount, type, and balance of proteins, carbohydrates, and fats of the diet greatly impact the gut microbiota [188], mainly due to the products from their degradation. This degradation causes the formation and release of SCFAs, phenols, indoles, and amines, with a wide range of physiological effects on the host [189]. 

It has been reported that a diet rich in prebiotics, such as inulin, oligofructose, and fructooligosaccharides, among other polysaccharides of plant origin, increase the growth of lactobacilli and bifidobacteria in gut microbiota [190]. Contrarily, the intake of a diet rich in simple carbohydrates and high in fat affects the abundance of Firmicutes phylum populations and decreases Bacteroidota, which, in some studies, has been related to obesity [191,192]. 

On the other hand, an animal-based diet is related to a greater microbial diversity increasing bile-tolerant bacteria and Bacteroidota, and decreasing Firmicutes [182]. The consumption of protein and fat of animal origin have also been linked to Bacteroidota phylum, while carbohydrates have been related to Bacteroidetes, as well as Firmicutes phylum, indicating an association with dietary patterns [193]. Additionally, the type of diet has an influence on intestinal transit time, which is faster with a plant-based diet than with an animal-based one [193,194]. Plant-based diets are a source of nondigestible fiber. It is generally accepted that the benefits of fiber intake on health are derived from laxation, increasing fecal bulking, and stool water content that stimulate mucus secretion and peristalsis [195].

Sprong et al. [196], in an in vivo study, observed a significant increase in the counts of bifidobacteria and lactobacilli when the diet included whey cheese or casein supplemented with threonine or cysteine. According to the aforementioned studies, gut microbiota composition can be positively modified through diet. Despite this, current food habits are characterized by no significant consumption of fruits, vegetables, and fish, leading to several health disorders, such as diabetes, hypertension, cardiovascular accidents, increased levels of cholesterol and triglycerides, greater insulin resistance, and inflammation, among others [197]. 

That is why there is an urgent need to develop efficient strategies aimed at reversing, preventing, and treating metabolic disorders associated with the dysbiosis process [198]. It can be achieved from a pharmacological and nutritional approach by incorporating prebiotics, probiotics, symbiotics, and other supplements into the diet. With their consumption, gut microbiota balance could be re-established or a healthy gut microbiota could be maintained when homeostasis has been lost due to an adverse condition [199].

### 4.1. Probiotics and Microbiota

The use of probiotic species has been relevant in the treatment of human and animal diseases due to their effect on the modulation of the intestinal microbiota. The potential that probiotic microorganisms represent has driven research for the production of probiotic foods and the modulation of gut microbiota, promoting their consumption [200].

The Food and Agriculture Organization and the World Health Organization define probiotics as strains of live microorganisms that confer beneficial effects on health when administered in specific amounts. The International Scientific Association for Probiotics and Prebiotics (ISAPP) supports this definition [201,202]. In recent years, a new terminology that provides a comprehensive approach to all the beneficial aspects of probiotics has been suggested. In this way, three main classes of probiotics have been proposed, including: (1) true probiotic (TP), to refer to a viable and active probiotic cell; (2) pseudo-probiotic (PP), which refers to a viable and inactive cell as a spore or vegetative body; and, (3) phantom probiotic (GP), to refer to a dead/nonviable cell that is intact or lysed [203]. According to this new terminology, a probiotic could be defined as a viable or nonviable microbial cell in a vegetative or spore state, intact or lysed, that is potentially healthy for the host [202].

Since the beginning of the 20th century, when the importance of the consumption of probiotic foods with a specific mixture of microorganisms began to be highlighted [199], studies on this type of food have been intensified in animal and human models. In this way, the beneficial effect of several species of specific strains, which have immunological, metabolic, and neuroendocrine activity, has been verified [199]. Certainly, the beneficial effects exerted by probiotic microorganisms are numerous, highlighting their effect on the development of microbiota that inhabits the organism (Table 1). Thus, it has been determined that the consumption of probiotic microorganisms helps to regulate intestinal homeostasis, maintaining an adequate balance between pathogens and bacteria necessary for the correct functioning of the organism [204].

Various genetic and molecular studies have made it possible to determine that probiotics exert beneficial effects through four main mechanisms (Figure 4): (1) the immunomodulation they exert in the host; (2) antagonism through the production of antimicrobial substances; (3) the inhibition of bacterial toxins; and (4) competition with pathogens by adhesion to the epithelium and by nutrients [216,217]. These mechanisms are relevant in the prophylaxis, treatment of infections, and maintenance of the host’s intestinal microbiota [202]. 

It is in the maintenance of the microbiota that probiotics play an important role in processes such as the absorption of cholesterol, the regulation of blood pressure, and glucose metabolism. [218,219,220]. In vivo studies have shown that the administration of *Bifidobacterium* and *Lactobacillus* strains has a significant impact on the composition of gut microbiota [221,222,223]. Park et al. [224] supplemented a diet with the probiotic strains *Lactilactobacillus cruvatus* HY7601 and *Lactiplantibacillus plantarum* KY1032, observing a decrease in the concentration of proinflammatory genes, greater expression of genes related to the oxidation of fatty acids in the liver, thus, as relevant alterations in the diversity and function of the microbiota. However, in clinical studies, this evidence is not conclusive, which is why it has been suggested that the observed changes may only occur in microbial metabolism at the level of SCFAs production and not in the microbial population [225].

Contrarily to these observations, several studies have demonstrated that probiotics produce changes in specific bacterial communities. Plaza-Díaz et al. [226] observed that the ingestion of *L. rhamnosus*, *Lactobacillus paracasei*, or *Bifidobacterium* induces changes in the adult fecal bacterial population. Similarly, it has been observed that the administration of *Lactobacillus salivarius* Ls-33 modifies the populations of fecal bacteria in adolescents with obesity, including several groups of clostridia [227]. On the other hand, in patients with irritable bowel syndrome, considerable fluctuations in clostridium populations have been observed. These alterations are related to the decrease in gastrointestinal symptoms after ingestion of *B. animalis* subsp. *lactis*, which is also related to the decrease in the concentration of pathogenic bacteria and the modification of the colonic production of SCFAs [212,228].

Probiotics play important roles when they come into contact with the rest of the microbial communities, influencing the metabolism of other members of the host’s microbiota [225]. An example of this is the bifidobacteria that metabolize a great diversity of carbohydrates, which come from the diet or the intestinal mucosa of the host and produce acetic and lactic acid in different proportions [229]. Specifically, *Bifidobacterium bifidum* increases its metabolism when it grows along with other species, such as *Bifidobacterium breve*, which enhances the catabolism of glycosylated compounds, such as mucin and 3-sialylactose [230,231].

Nevertheless, a modification of the intestinal microbiota through the consumption of probiotics is not the only way to produce a beneficial effect in the host. The probiotic effect can be manifested through the interaction with the immune system [232], but it is necessary to highlight the challenges and opportunities regarding the studies of probiotic microorganisms capable of generating long-term effects after modifying the intestinal microbiota. In this context, in addition to traditional health-promoting bacteria (*Bifidobacterium* and lactic acid bacteria), in recent years, the beneficial effect has been noted after the therapeutic use of next-generation probiotics [233]. This concept includes microorganisms such as *Akkermansia muciniphila* whose effect is associated with glucose metabolism, lipid metabolism, and intestinal immunity [234], for which it has been proposed as a target for immunotherapy in various types of cancer [233,235].

Recently, some authors have proved the benefits of consuming probiotics through clinical studies. Hou et al. [236] determined beneficial changes in the gut microbiota of healthy adults after the consumption of *Lactobacillus casei* Zhang. The effect was related to enterotypic changes due to the increment in concentration of beneficial microorganisms, inhibiting the growth of pathogenic ones. Such changes increased both the development of lactobacilli and the beneficial metabolic functions of gut microbiota. Similarly, the consumption of probiotic fermented milk and yogurt by healthy adults has shown a direct effect on the concentration changes of *Bifidobacterium* spp., especially *B. longum* [237]. The consumption of probiotic fermented milk also has proved to generate beneficial changes in obese patients [238]. Likewise, the inclusion of a mixture of eight probiotic bacteria (*Lactobacillus* spp., *Bifidobacterium*, and *Streptococcus thermophilus*) into the diet of obese people during 15 days generates changes in the concentration of bifidobacteria. The main effect is related to a decrease in *Collinisela*, a proinflammatory biomarker, and an increase in *Akkermansia* concentration, as well as an improvement in oxidative stress biomarkers [239]. Probiotic consumption, specifically *L. casei*, could improve the management of diarrhea in infants, affecting both the modification of gut microbiota and the attenuation of inflammatory biomarkers [240].

Pharmacological treatments should not be replaced by probiotics, but their consumption could be incorporated into the diet during disease management to provide their well-documented beneficial effects. Thus, the development of probiotic foods containing vehicles for microorganisms that exert a benefit in the modulation of gut microbiota is an area of opportunity in the food science and technology field [241]. Of course, one of the most important challenges is to guarantee the survival of probiotic microorganisms in sufficient concentrations to reach the adequate amounts that promote changes in metabolic functions of gut microbiota.

### 4.2. Prebiotics and Microbiota

Prebiotics are currently defined by ISAPP as “a substrate that is selectively utilized by host microorganisms conferring a health benefit” [242]. Among the best-known prebiotics are inulin, fructooligosaccharides, galactooligosaccharides, and lactulose [243]. These molecules are selectively fermented in the colon, conferring beneficial effects to the host, including stimulation of both growth and metabolic activity of various bacterial groups of the intestinal microbiota [244], mainly probiotics, which use prebiotics as a carbon source [242,245]. 

Recently, another type of compound including polyphenols has been proposed as a prebiotic since they meet the current definition previously mentioned. It appears that their beneficial effect on the host depends on the microbial utilization and the metabolites produced rather than on parent compounds. However, as studies on these emerging prebiotics are not yet conclusive, more evidence on the health benefits associated with polyphenols and probiotic interaction is needed [245].

Bacterial fermentation of prebiotics leads to the production of SCFAs, mainly butyrate, acetate, and propionate [246]. These acids have an impact on various cellular mechanisms, such as the activation of G-protein-coupled receptors and the inhibition of histone deacetylation [247]. Likewise, they act as a source of energy for colorectal tissues, exert an anti-inflammatory effect, and act as molecules that are related to the signaling pathways of the microbiota–intestine–brain axis [248]. In addition, other organic acids, such as formate, lactate, and succinate, decrease intestinal pH, preventing the growth of pathogenic bacteria [249]. 

Thus, it has been indicated that the consumption of prebiotics by healthy adults increases the concentration of *Bifidobacterium* spp. and *Lactobacillus* spp. in gut microbiota [250]. However, the composition of the microbiota after the consumption of certain prebiotics is not limited only to these taxa. Through the application of sequencing techniques and metagenomic analysis, it has been shown that prebiotics affect the entire composition of gut microbiota (Table 2) [251,252,253].

On the other hand, studies also suggest that beneficial changes in gut microbiota are maintained with continuous consumption of prebiotics. In addition, microbial diversification is dependent on the basal or indigenous microbiota, including the growth of specific species and the enzymatic capacity of some strains [248,261]. For example, resistant starch is considered a prebiotic, since it can be fermented in the colon, conferring beneficial metabolic effects. Among the most studied benefits, the following stand out: an increase in the turnover of bile salts, laxative effect, control of blood lipid levels, and a decrease in the postprandial glucose response [262]. In addition, this carbohydrate contributes to cell growth and proliferation by increasing the concentration of butyrate once it has been fermented in the intestine [263]. 

The metabolic effects, as well as the group of bacteria that are favored after the incorporation of starch, depending on the type of resistant starch that is consumed [264]. For example, type 4 resistant starches have been found to favor the growth of *Bacteroides* and *Parabacteroides* spp. in the intestine [256]. On the other hand, type 2 resistant starches increase the populations of *Ruminococcus bromii* and *Eubacterium rectole* spp. in humans [258], and *Bifidobacterium*, *Akkermansia*, and genera of *Allobaculum* in murine models [259]. Regarding type 3 resistant starches, several studies using animal and human models have shown that they favor the growth of beneficial bacterial populations, mainly SCFA-producing genera, such as *Prevotella*, *Ruminococcus*, *Lachnospiraceae*, *Veillonellaceae*, *Bulleidia,* and *Dialister* [257].

Alginate is another polysaccharide that has stood out for its prebiotic properties. Its consumption modifies the intestinal microbiota by increasing the relative abundance of microbial populations such as *Roseburia*, *Ruminococcus*, and *Lachnospira*, which are SCFA producers. Additionally, an increase in the concentration of bifidobacteria due to the consumption of alginate has been reported [257].

In the same way, an increase in the concentration of *Prevotella* and *Roseburia* has been observed by the in vitro fermentation of prebiotics such as inulin, galacto-oligosaccharides (GOS), and xylo-oligosaccharides (XOS) of corn and sugar cane with a high content of fiber and oat β-glucans. Furthermore, there is a concomitant increase in propionate production. Similarly, inulin, XOS, and GOS have a strong bifidogenic effect on the microbial composition and are precursors of the formation of butyrate by native microorganisms of the intestinal microbiota [254].

Other studies highlight the effects of barley β-glucans, which, when administered in low doses, significantly increase the count of bifidobacteria and lactobacilli in healthy subjects. Similarly, β-glucans from wheat flour and whole barley pasta increase the levels of *Roseburia hominis*, *Clostridium orbiscindens*, *Clostridium* sp., and *Ruminococcus* sp. in the gut microbiota. At the same time, a reduction in the levels of Firmicutes and Fusobacteria is observed, and it has been verified that the consumption of β-glucans affects the increase in the concentration of acids such as 2-methylpropanoic, acetic, butyric, and propionic [259].

In addition to prebiotic sources of plant origin, the effect of bovine milk oligosaccharides has also been evaluated, which, in combination with GOS, decrease the concentration of *Clostridium perfringens*, with a simultaneous increase in Bifidobacteria, lactate, and acetate [255]. Despite the results, some authors suggest that prebiotic supplementation does not always lead to a global change in the alpha or beta diversity of gut microbiota. However, a modification is observed in the abundance of certain bacterial taxa, such as *Ruminococcaceae* (*Clostridium cluster* IV), *Parabacteroides*, and *Phascolarctobacterium* [265].

Currently, the role of FODMAPs in the intestinal microbiota has been studied, noting that the low intake of these compounds is related to a reduction in the symptoms of various gastrointestinal diseases, including irritable bowel syndrome and inflammatory bowel disease [266,267]. Conversely, a diet high in FODMAPs seems to have an opposite effect to prebiotic supplementation, decreasing *Bifidobacterium* populations and increasing bacteria associated with dysbiosis [266]. However, more studies are needed to evaluate the effects of a diet high in FODMAPs on the composition of gut microbiota, focusing on the persistence of changes in microbial composition and adverse health effects.

## 5. Conclusions

The intestinal microbiota has been considered as another organ of the human body, with its own characteristics that make it essential in metabolic functioning. Due to these characteristics, dysbiosis has been implicated as one of the main factors in the development of diseases such as diabetes, obesity, cancer, or those related with neurodegenerative problems. Thus, the maintenance of the correct balance in the composition of taxa that form gut microbiota is of utmost importance. This review fulfills the objective of informing about the importance of the intestinal microbiota for the human’s good health. At the same time, the consumption of probiotics and prebiotics directly affects both the maintenance of the composition’s balance of microbiota and the prevention/management of dysbiosis. However, despite the studies carried out in this field, they are still not conclusive in the direct relationship of microbiota, such as the Firmicutes/Bacteroidota, in the development of diseases. This is clearly an area of opportunities and challenges that food science research should consider in order to determine in a conclusive way the role that diet plays in maintaining the good composition of microbiota and, likewise, the role that it plays within the health–disease state of the human.

## Figures and Tables

**Figure 1 foods-10-01261-f001:**
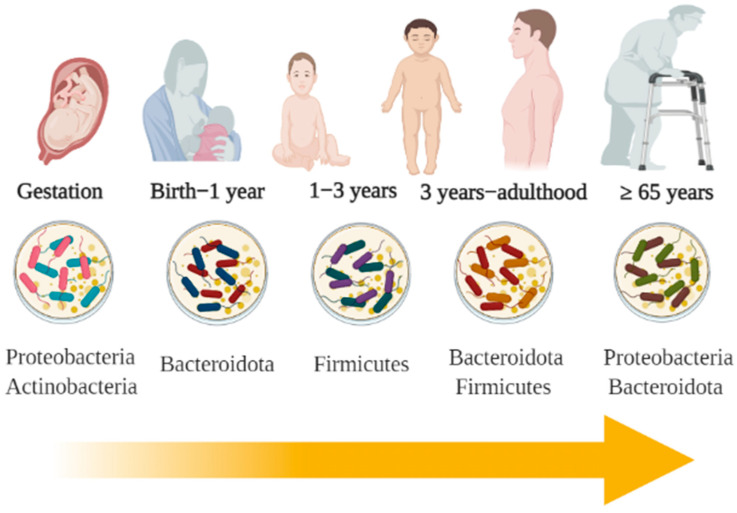
Dominant gut microbiota phyla in different life stages.

**Figure 2 foods-10-01261-f002:**
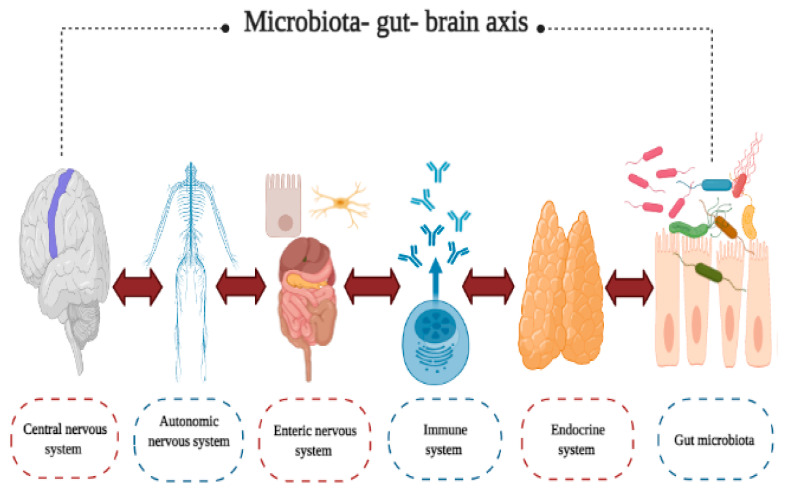
Microbiota–gut–brain axis, bidirectional communication network that includes the central nervous, autonomic nervous, enteric nervous, immune, and endocrine systems, and gut microbiota.

**Figure 3 foods-10-01261-f003:**
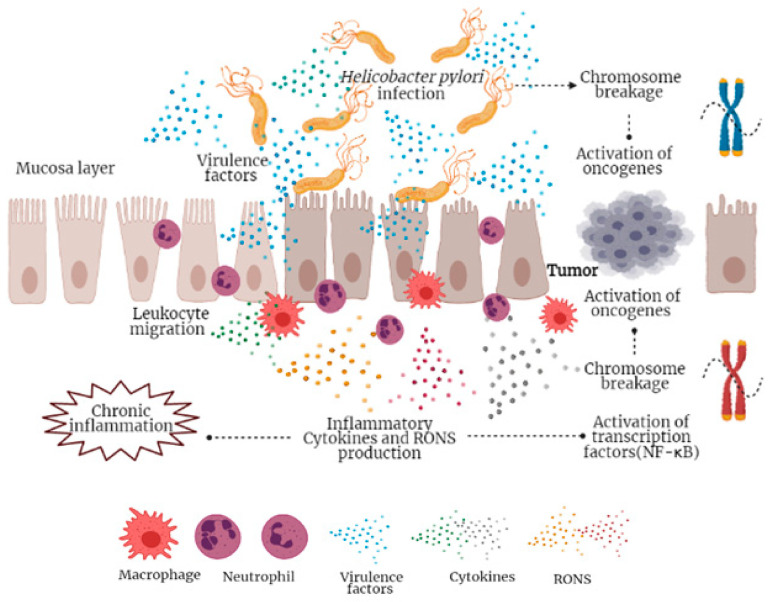
Mechanisms in the pathogenesis of *Helicobacter pylori*-associated gastric carcinogenesis.

**Figure 4 foods-10-01261-f004:**
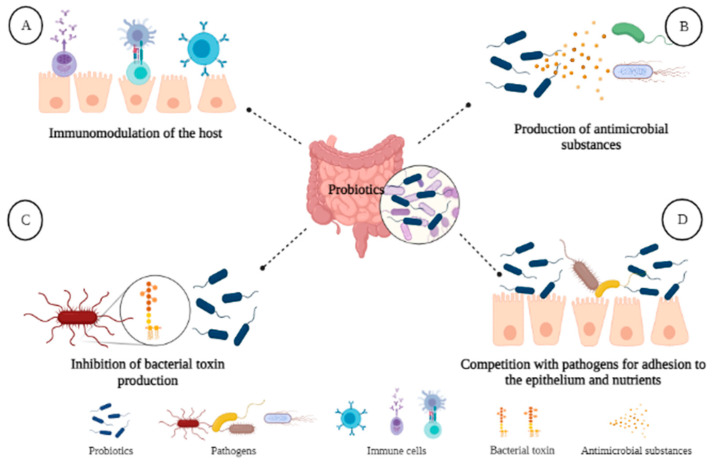
Action mechanisms of probiotics during gut colonization. (**A**) Human immunomodulation exerted by probiotics; (**B**) antagonism derived from production of antimicrobial substances; (**C**) inhibition of toxin action by probiotic metabolism; (**D**) control of pathogen adhesion on epithelium by a competition mechanism.

**Table 1 foods-10-01261-t001:** Probiotics and effects on gut microbiota.

Probiotic Strain	Model Description	Effects on Gut Microbiota	Source
*Lactobacillus helveticus* Bar13	Healthy male and female adults aged between 71 and 88 years. Consumption of 10^9^ CFU/mL of *L. helveticus* Bar13 once a day for 30 days	No increase in clostridium cluster XI	[205]
*Lactobacillus rhamnosus GG*	Healthy adults (mean age of 42 years). Daily consumption of 250 mL milk-based fruit drink containing either *L. rhamnosus* GG (ATCC 53103, 6.2 × 10^7^ CFU/mL) for three weeks	No significant impact on gut microbiota composition	[206]
*Lactobacillus casei* Zhang	Healthy adults consuming 10^6^ CFU/mL of *L. paracasei* Zhang for 28 days	Difference in composition and diversity of intestinal microbiota compared to baseline	[207]
*Lactobacillus paracasei* DG	Healthy male and female adults aged between 23 and 55 years. The subjects consumed a capsule containing at least 24 billion viable *L. paracasei* DG cells for 4 weeks	Increase in proteobacteria and *Coprococcus*; decrease in *Blautia*	[208]
*Lactobacillus salivarius UBLS22*	Healthy young volunteers. Consumption of 2 × 10^9^ CFU/mL of *L. salivarius* for 6 weeks	Increase in lactobacilli and decrease in *E. coli*	[209]
*Lactobacillus casei* NCDC 19	Male C57BL/6 mice (6–7 weeks old). Diet supplemented with 10^8^ CFU/mL of *L. casei* NCDC 19 for 8 weeks	Increase in bifidobacteria population	[210]
*Bifidobacterium animalis* subsp. *Lactis*	Adult female monozygotic twin pairs consuming 4.9 × 10^7^ CFU/mL of *B. animalis* subsp. *Lactis* for 7 weeks	No change in dominant microbiota	[211]
	Subjects with irritable bowel síndrome consuming 10^9^ CFU/mL of *B. animalis* subsp. *Lactis* for 4 weeks	Increase in butyrate producing species and decrease in a pathobiont, *Bilophila wadsworthia,* abundance	[212]
*Akkemansia muciniphila*	Obese and Type 2 Diabetic mice fed with 2 × 10^8^ CFU/0.2 mL of *A. muciniphila* for 4 weeks	Increase in the intestinal endocannabinoid content (acylglycerols)	[213]
*Faecalibacterium prausnitzii*	Adult male Sprague–Dawley (SD) rats with colitis consuming 10^9^ CFU/mL of *F. prausnitzii* for 7 days	Induction of interleukin IL-10 production	[214]
*Bacteroides uniformis* CECT 7771	Obese adult male wild-type C57BL-6 mice consuming 5.0 × 10^8^ CFU/mL of *B. uniformis* CECT 7771 for 7 weeks	Partial stabilization of the microbiota	[215]

**Table 2 foods-10-01261-t002:** Prebiotics and their effects on gut microbiota.

Prebiotic	Model description	Effects on gut microbiota	Source
Galacto-oligosaccharides (GOS)	Healthy adult volunteers. GOS daily dose of 2.4 g	Increase in *Bifidobacterium* and *Lactobacillus* abundance	[250]
Alpha-GOS, betha-GOS, Xylo-oligosaccharides (XOS), β-glucan, inuline	In vitro fermentation of standardized fecal sample from healthy adult volunteers. Positive control: 4 mg/mL of fructooligosaccharides. 5 different concentrations of each prebiotic assayed	β-glucan: Increase in Bacteroidotes (*Prevotella*) and Firmicutes (*Roseburia*) Alpha-GOS and XOS: Increase in bifidobacteria	[254]
Bovine milk oligosaccharides (BMO)	BMO and lactose co-culture effect on *Bifidobacterium longum* subsp. *longum* metabolism and *Clostridium perfringens* inhibition	Decrease in *Clostridium perfringers* and increase in *Bifidobacterium*	[255]
Resistant starch type 4 (RST4)	Subjects with metabolic syndrome. 26 weeks treatment including two 12-week intervention periods, one for RST4 (30%, *v/v* in flour) and one for control flour	Increase mainly in *Bacteroides* and *Parabacteroides* spp. Microbial enrichment involved *Christensenella minuta,* recently identified in human feces	[256]
Resistant starch type 3	Pigs, 3 months old. The percentage of the prebiotic diet was increased in 20% increments until 100% was reached	Increase in *Prevotella*, *Ruminococcus*, and *Lachnospiraceae*	[257]
Resistant starch type 2 (RST2)	Male C57BL/6J mice (18–20 months old)	Increase in *Ruminococcus bromii*, *Eubacterium rectale*	[258]
	Healthy male and female human subjects aged between 23 and 38 years. RST2 daily dosis of 100 g for 2 weeks	*Bifidobacterium* spp., *Allemansia*, and *Allobacum* genera	[259]
β-glucan	Healthy human subjects. Barley β-glucans daily dose of 3 g for 2 months	Increase in *Prevotella*, *Roseburia*, and *Clostridium*, decrease in Firmicutes and Fusubacterium concentration	[260]

## Data Availability

Not applicable.

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
