# Peer review of "Impact of the Gut Microbiota Balance on the Health–Disease Relationship: The Importance of Consuming Probiotics and Prebiotics"

_foods, 2021, doi:10.3390/foods10061261_

Round 1

Reviewer 1 Report

The aim of the review falls within the thematic scope of the journal.

The review discusses the current state of knowledge about the relationship between the consumption of probiotics and prebiotics and the balance of the gut microflora, and the importance of the gut microflora in the emergence and development of chronic and degenerative diseases.

The topic is very interesting, very broadly and thoroughly treated by the authors. A perfectly conducted review of the current literature on the subject.

All remarks (lines 122, 496, 545, 570, 689, 772, and 783) are marked in the text of manuscript in the review mode. After introducing these minor changes, in my opinion, the Editors may direct the manuscript for further processing.

Author Response

Point 1. All remarks (lines 122, 496, 545, 570, 689, 772, and 783) are marked in the text of manuscript in the review mode. After introducing these minor changes, in my opinion, the Editors may direct the manuscript for further processing.

Answer. All remarks were reviewed and corrected

Reviewer 2 Report

The authors review various diet-microbiota-disease interactions. Many reviews cover the same material but this paper provides particular detail on the microbiota-cancer relationship.

The manuscript contains multiple English errors that require correction by an expert.

Specific comments:

Pylori is misspelled in figure 3.

Italicize bacteria names and use lower-case.

Line 458: include citations after 1)

Lines 515-518 and 532-535: list using numbered parentheses separated by ;

Highlights in the manuscript (attached) indicate English errors (spelling, punctuation, sentence structure, word choice, etc.).

Once a word/phrase is abbreviated and defined, use the abbreviation consistently.

Table 1: if this includes human, animal and in vitro experiments, include this information.

Figure 4: Each panel (A, B...) needs to be described in the legend. Define the images in the figure, as you do in Figure 3.

Line 662: irritable bowel syndrome

Author Response

Point 1. Pylori is misspelled in figure 3.

Answer. The word Pylori was written correctly in figure 3. It was also revised in all the manuscript and it was corrected when needed

Point 2. Italicize bacteria names and use lower-case.

Answer. Bacteria names were italicized in all the text

Point 3. Line 458: include citations after 1)

Answer.  The corresponding citation after 1) was included

Point 4. Lines 515-518 and 532-535: list using numbered parentheses separated by ;

Answer. The text was changed according reviewer suggestion 

Point 5. Highlights in the manuscript (attached) indicate English errors (spelling, punctuation, sentence structure, word choice, etc.).

Answer. English errors highlighted were revised and corrected

Point 6. Once a word/phrase is abbreviated and defined, use the abbreviation consistently.

Answer. The abbreviations used in the manuscript were checked and we have used them consistently through the text

Point 7. Table 1: if this includes human, animal and in vitro experiments, include this information.

Answer. Table 1 was completed with the information required 

Point 8. Figure 4: Each panel (A, B...) needs to be described in the legend. Define the images in the figure, as you do in Figure 3.

Answer. Figure 4 was modified describing each panel in the legend as well the images used in the figure

Point 9. Line 662: irritable bowel syndrome

Answer. Intestinal irritable syndrome was changed to irritable bowel syndrome

Reviewer 3 Report

Please see the attached file for specific comments. 

You are advised to restructure some sections to facilitate coherence. Also in several points, regardless the detailed research of previous studies and the presentation of results, the authors are suggested to present  what is the "take-away" message that the readers should keep. What are the challenges, the opportunities or future research studies?

Author Response

Point 1. Lines 26-27, Abstract, “The gut microbiota..tract”. General, it is suggested to remove the article “the” before gut microbiota, in most places in the manuscript.

Answer. “The gut microbiota” was changed to “gut microbiota” in most part of the text 

Point 2. Line 43: “phylos”, the plural or phylum is “phyla”, please modify throughout the manuscript where appropriate

Answer. Phylos has been modified to its correct form in all the text

Point 3. Line 136 “progressively follows by various microbial strains.”. Please modify

Answer. The phrase was modified to “After birth, the intestine is progressively colonized by various microbial strains”

Point 4. Line 642: “in vitro” should be written with italics throughout the text. 

Answer. “In vitro” was italicized in all the text 

Point 5. Lines 72-77: The authors are suggested to better enhance the contribution of the current review paper compared to previous studies.

Answer. The contribution of this review was improved 

Point 6. Also lines 53-71 could be modified and restructured to better confer how gut microbiota relates to disease development. For instance, lines 57-58, imply that manipulation of gut microbiota is seldom successful, a statement that is not very valid.

Answer. Following your suggestion, paragraph was restructured

Point 7. Line 91, “these compounds (SCFA)...” Do all of the produced SCFA relate to these benefits? Potentially, the beneficial activity of each specific fatty acid could be shortly mentioned (either here or later in the text as the authors will notice in a following comment).

Answer. According Douglas J. Morrison & Tom Preston 2016, some beneficial activity of each SCFA has been mentioned

Point 8. Please note for Figures if other studies were used, for instance Figure 2 (similar) and Figure 3 seem quite similar to other studies.

Answer.  Although the style of the figures presented in this manuscript is similar to those from other studies, the information and design are ours. Figure 2 and 3 present summarized information from the review carried out

Point 9. Line 482-486: Does the interaction and the synergistic effects of different microbial species and their metabolites get affected by diet? For instance SCFA produced by one species might be consumed by other bacterial species, and these could be affected by diet?

Answer. 

The answer to your questions is yes.

It is well known that diet has a direct effect on the microbial species of gut microbiota, as it has been widely proven that prebiotics selectively stimulates the growth of beneficial species such as bifidobacteria and lactobacilli. Regarding your second question, it has been reported that products of prebiotic fermentation by lactobacilli and bifidobacteria, such as lactic and acetic acid, can be subsequently degraded by other bacteria such as Anaerostipes caccae or Roseburia intestinalis. Since prebiotics are more abundant in plant-based products, the diet has a direct effect because it strongly influences the growth of certain types of microbial species and their metabolites.

Saulnier et al., 2009. https://doi.org/10.1016/j.copbio.2009.01.002

Point 10. Lines 489-490: “with this type of diet than with those rich in animal fat and protein” Briefly explain the reason for that.

Answer. The effect of insoluble fiber from a plant-based diet on gastrointestinal transit was briefly explained and the corresponding citation has been added

Point 11. Lines 487-494: The introducing sentence of this paragraph, does not coincide with the rest..The authors start from one point, move to another and do not facilitate coherence.

Answer. The paragraph was rewritten to be more consistent

Point 12. Lines 497, and 499, there are two references (194 and 181) but the way text is written, seems to refer to one study. Please check and modify if necessary.

Answer. The paragraph has been checked and modified

Point 13. Line 505: What do the authors mean by “drug-nutritional approach”?

Answer. We meant using both supplements and diet to enhance probiotics, prebiotics, and symbiotic consumption through diet. To avoid confusion the paragraph was rewritten.

Point 14. Section 4.1 Probiotics and microbiota. However, in serious disease we should make it clear that probiotics and prebiotics can’t replace medicine, but are a means to modify gut microbiota and mainly prevent the development of disease.

Answer. Information related to your comment was included in section 4.1

Point 15. Lines 514-525: Does ISAPP supports these different types-kinds of probiotics?

Answer. After an exhaustive search in the scientific literature, we did not find any ISAPP document supporting the types of probiotics mentioned in the text, but it should be taken into account that this classification was only proposed last year by Zendeboodi et al. in their article “Probiotic: Conceptualization from a New Approach” doi:10.1016/j.cofs.2020.03.009. 

Point 16.  Table 1: should be restructured and the column with the reference study should be moved to the right side.
 Further detail on the type of the study would be also useful, as this Table is quite general and does not facilitate the readers.

Answer. Table 1 was restructured according to your suggestion. Besides, information about the type of study conducted was added

Point 17. Line 584: This section (4.1) does not add much as It is well documented that gut microbiota is affected by probiotics. Since this section refers to the prevention of disease through probiotics administration, then the authors could modify it accordingly and present some more targeted examples, particularly with clinical studies and human trials, were the probiotic had positive effects on health markers through gut microbiota modifications. 

Answer. 

The positive effects of probiotics consumption on health markers through gut microbiota modifications reported in some recent clinical studies carried out in humans have been included in section 4.1:

Redondo-Useros et al., 2019 doi:10.3390/nu11030651

Hibberd et al, 2019  DOI 10.3920/BM2018.0028

Cancello et al. 2019 doi:10.3390/nu11123011

Lai et al., 2019 doi:10.3390/nu11051150

Hou et al., 2020 https://doi.org/10.1080/19490976.2020.1736974

Point 18. Again however, it should be made clear to the reader, that probiotics are not drugs.

Answer. It has been pointed out in the text that probiotics are not drugs

Point 19. Line 593: “indigestible elements” does not seem scientifically “sound”.

Answer. The definition was changed according to the current ISAPP consensus

Point 20. Lines 593-600: Recent ISAPP meeting in 2017 concluded in an updated prebiotic definition that also includes other compounds apart from carbohydrates, like polyphenols.

Answer. It has been included a brief paragraph mentioning the recent proposal of polyphenols as prebiotics

Point 21. Lines 601-608: All of these acids have the same effect? The authors could also refer to more specific examples, for instance butyrate has been shown to reduce appetite.

Answer. Some specific beneficial activity of each SCFA have been mentioned before this paragraph

Point 22. Table 2: Similar observations here for the format, as with Table 1.
 Also some more detail would be useful, more specifically if the study refers to in vitro experiments or clinical trials.

Answer. Table 2 format was changed and information about the type of study (in vitro experiments or clinical trials) was included

Point 23. General comment: Both section on prebiotics and probiotics, refer to previous studies demonstrating the outcomes, but at the end of each section it is advised that the authors confer the “take-away” message to the reader..What is the current challenge? How should we address the consumers?

Answer. A closing paragraph containing some information related to your questions was added

Point 24. The following references could be also included
Terpou et al., Probiotics in Food Systems: Significance and Emerging Strategies Towards Improved Viability and Delivery of Enhanced Beneficial Value. Nutrients. 2019 11(7):1591. 

Answer. The reference suggested was added and it was also used to describe challenges and opportunities in the field of gut microbiota balance by probiotics 

Round 2

Reviewer 2 Report

The authors have made substantial improvements to the paper. Below are remaining corrections suggested:

Line 166 "are related"

176 - TNF (use correct alpha sign)

286 - showed

648 - through the application of...

Table 2: RST4 - metabolic syndrome

B-glucan: daily dose

719 - directly affects both the maintenance of the composition’s balance of microbiota and the prevention/management of dysbiosis.

Author Response

Line 166 "are related"

It was corrected

176 - TNF (use correct alpha sign)

Sign was corrected

286 - showed

Verb conjugation was corrected

648 - through the application of...

The phrase was changed

Table 2: RST4 - metabolic syndrome

It was corrected

B-glucan: daily dose

It was corrected

719 - directly affects both the maintenance of the composition’s balance of microbiota and the prevention/management of dysbiosis.

The phrase was changed according reviewer's suggestion

Reviewer 3 Report

In the revised version of the manuscript the authors have now addressed all the comments and the suggestions raised by the reviewers. The manuscript can proceed to publication prior to some minor English modifications and spell checking in the newly inserted parts (e.g. Lines 631-632, in vivo should be also italicised).